# Dispersion and Polishing Mechanism of a Novel CeO_2_-LaOF-Based Chemical Mechanical Polishing Slurry for Quartz Glass

**DOI:** 10.3390/ma16031148

**Published:** 2023-01-29

**Authors:** Zifeng Zhao, Zhenyu Zhang, Chunjing Shi, Junyuan Feng, Xuye Zhuang, Li Li, Fanning Meng, Haodong Li, Zihang Xue, Dongdong Liu

**Affiliations:** 1Key Laboratory for Precision and Non-Traditional Machining Technology of Ministry of Education, Dalian University of Technology, Dalian 116024, China; 2School of Mechanical Engineering, Hangzhou Dianzi University, Hangzhou 310018, China; 3School of Mechanical Engineering, Shandong University of Technology, Zibo 255000, China

**Keywords:** quartz glass, chemical mechanical polishing, CeO_2_ composite abrasive, environment friendly

## Abstract

Quartz glass shows superior physicochemical properties and is used in modern high technology. Due to its hard and brittle characteristics, traditional polishing slurry mostly uses strong acid, strong alkali, and potent corrosive additives, which cause environmental pollution. Furthermore, the degree of damage reduces service performance of the parts due to the excessive corrosion. Therefore, a novel quartz glass green and efficient non-damaging chemical mechanical polishing slurry was developed, consisting of cerium oxide (CeO_2_), Lanthanum oxyfluoride (LaOF), potassium pyrophosphate (K_4_P_2_O_7_), sodium N-lauroyl sarcosinate (SNLS), and sodium polyacrylate (PAAS). Among them, LaOF abrasive showed hexahedral morphology, which increased the cutting sites and uniformed the load. The polishing slurry was maintained by two anionic dispersants, namely SNLS and PAAS, to maintain the suspension stability of the slurry, which makes the abrasive in the slurry have a more uniform particle size and a smoother sample surface after polishing. After the orthogonal test, a surface roughness (S_a_) of 0.23 nm was obtained in the range of 50 × 50 μm^2^, which was lower than the current industry rating of 0.9 nm, and obtained a material removal rate (MRR) of 530.52 nm/min.

## 1. Introduction

Quartz glass is an amorphous form of silicon dioxide (SiO_2_) [1]. Because of its excellent physical and chemical properties, SiO_2_ is widely used in military, aerospace, chemical, and optical lens applications [2]. However, due to the low fracture toughness [3], the machining process of quartz glass leads to defects such as chipping, cracking, scratching, and sub-surfacing [4]. To meet the high surface-quality requirement of optical components, processing techniques such as magneto-rheological polishing (MRF) [5,6], jet polishing (JP) [7], ion beam polishing (IBP) [8,9], and chemical mechanical polishing (CMP) [10,11,12] are preferred. CMP can effectively reduce surface roughness (S_a_) while eliminating sub-surface damage and has become preferable in the wafer processing industry due to its relatively low cost and simple operation.

During the CMP process [13,14], a softened layer is formed by the chemicals in the polishing slurry on the sample surface. The chemicals are then mechanically removed by soft abrasives to obtain a scratch-free and ultra-smooth surface. The chemical etching process is primarily achieved by using chemicals with strong corrosion and toxicity, such as potassium hydroxide (KOH) [15,16], phenol [17], ammonium hydrogen fluoride [18], and sodium dodecylbenzene sulfonate (SDBS) [19]. For example, a SiO_2_-based CMP slurry was developed, using Polyvinylpyrrolidone (PVP) as the dispersant, citric acid as the complexing agent, guanidine carbonate as the co-solvent, and KOH [16] to adjust the pH to 11. The polishing tests of quartz glass yielded a material removal rate (MRR) of 169.50 nm/min and a S_a_ of 0.73 nm. Quartz glass polishing was performed by controlling the degree of aggregation of colloidal cerium dioxide (CeO_2_) using KOH [15]. A S_a_ of 0.20 nm was obtained. Polishing tests of the SiO_2_ wafer were conducted using an acidic SiO_2_ polishing slurry with a phenol [17] addition. A S_a_ of 0.19 nm was obtained, along with a MRR of 181.70 nm/min. Quartz glass wafers were also polished with slurry that used ammonium bicarbonate fluoride and hydrogen fluoride (HF) to adjust the pH. The S_a_ was reduced to ~100 nm, and the visible light transmission reached up to 89% [18]. Though an excellent surface quality was obtained after polishing in the above study, a greener and more environment friendly chemical mechanical polishing slurry, which has comparable polishing performance, is needed for industrial application.

Except for the conventional study of the chemical composition, slurry dispersion has also been a key factor in CMP. A dispersion system configured with the compound of SDBS and PVP exhibited excellent dispersion performance, and the slurry did not settle for 72 h. However, the S_a_ of quartz glass after polishing using such polishing slurry was ~10 nm. Moreover, it is still unknown how the dispersion performance of the slurry influences the polish performance.

In this study, a novel cerium oxide (CeO_2_)-Lanthanum oxyfluoride (LaOF)-based composite abrasive polishing slurry was developed based on quartz glass polishing. The polishing slurry consisted of CeO_2_, Lanthanum oxyfluoride (LaOF), potassium pyrophosphate (K_4_P_2_O_7_), sodium N-lauroyl sarcosinate (SNLS), and sodium polyacrylate (PAAS). SNLS and PAAS formed the dispersion system. Through orthogonal experiments, the best polishing parameters and polishing slurry composition ratios were obtained. The S_a_ of 0.23 nm was obtained in the 50 × 50 μm^2^, which was much lower than the S_a_ of 0.9 nm S_a_ from the commercial slurry. The MRR was as high as 530.52 nm/min, 77.34% higher than the traditional pure CeO_2_-based polishing slurry, while the S_a_ was reduced by 31.04%. The slurry dispersion under different dispersant concentrations was studied using a laser particle sizer, UV absorber, and viscometer. Finally, the CMP mechanism of the presented polishing slurry was analyzed based on X-ray photoelectron spectroscopy (XPS) and Fourier infrared spectroscopy (FTIR).

## 2. Materials and Methods

### 2.1. Experimental Materials

The 99.99% pure quartz glass sample pieces (diameter 10 mm, thickness 3 mm) were purchased from Guanghe Quartz Products Co., Ltd., Lianyungang, China. SNLS, PAAS, and K_4_P_2_O_7_ were purchased from Maclean’s Chemical Reagents, with an analytical purity of 98%. CeO_2_ and LaOF were purchased from Inner Mongolia Guangheyuan Nanotechnology Co, Ordos, China.

### 2.2. Configuration of Polishing Slurry

The polishing slurry was configured by 0.5 wt% of rare earth abrasives CeO_2_ and LaOF, 0.5 wt% of dispersants SNLS and PAAS, and the pH was adjusted by K_4_P_2_O_7_ to 9.5. The polishing slurry was configured with OS20-S mechanical stirrer, which came from Beijing Dalong Xingchuang Experimental Instruments JSC, Beijing, China; the OPBM-2 ball mill was used to disperse the polishing slurry, which came from Shenzhen Jitong Technology Development Ltd., Shenzhen, China with zirconium balls as the grinding media. Ultrasonic assisted dispersion with YS0615 ultrasonic cleaner for 30 minutes before polishing, equipment from Shenzhen Yunyi Technology Ltd., Shenzhen, China. An HSC-19T magnetic stirrer was used for continuous stirring during the polishing process, with equipment from Qunan Experimental Instruments (JOANLAB) Ltd., Huzhou, China. After polishing, the polished samples were rinsed with deionized (DI) water and dried by compressed air.

### 2.3. CMP Tests

Quartz glass sample pieces were ground using a #1500 diamond abrasive disc and DI water as a grinding solution. This process removed the surface damage caused by the previous machining process on the quartz glass surface and saved time for subsequent CMP processing. The S_a_ of the quartz glass sample was 22.25 nm after grinding, and the measured area was 100 × 100 μm^2^. Grinding and polishing are performed using UNIPOL-1200S precision polishing machine from Shenyang Kejing Automation Equipment Ltd., Shenyang, China. The grinding and polishing slurry is supplied using SKZD-4 drip irrigation machine, equipment from Shenyang Kejing Automation Equipment Ltd., Shenyang, China. Polishing tests were performed using a polyurethane polishing pad with a polishing pressure of 40 kPa, a polishing speed of 140 rpm, a polishing slurry flow rate of 2 mL/min, and a polishing time of 20 min. 

To obtain the optimal combination of polishing slurry and the polishing process parameters, six factors were tested orthogonally in this test, namely: abrasive concentration, polishing slurry pH, dispersant concentration, pressure, speed, and flow rate of the slurry, as shown in Table 1. For the polishing slurry dispersant concentration, comparative experiments were conducted to study the influence on dispersibility.

The S_a_ and MRR were selected as the evaluation indices for polishing performance. The equation for calculating MRR can be expressed as:(1)$$\mathrm{MRR}=\frac{\Delta m}{\rho s\tau }{\;\times \;10}^{7}$$
where ∆m (g) is the mass difference before and after polishing; ρ is the quartz glass density of 2.2 (g/cm^3^); s (cm^2^) is the surface area of the sample piece to be polished; and τ (min) is the polishing time.

### 2.4. Characterization

The grinding morphology of the polished slurry was examined using a TESCAN MIRA scanning electron microscope at an accelerating voltage of 20 kV. The HITACHI HT7700 transmission electron microscope was used to detect the thickness of the damaged layer before and after polishing of the sample, with an ac-acceleration voltage of 100 kV. The S_a_ of the samples before and after polishing was measured using an Olympus MX40 optical microscope. A Zygo NewView^TM^ 9000 white light interferometer was used to measure the roughness of the sample surface before and after polishing. The difference in mass before and after polishing was weighed with an EX224ZH analytical balance to calculate the MRR.

The properties of the polishing slurry were analyzed in terms of particle size and zeta potential. The tests were performed with a Zetasizer Pro particle size analyzer, equipment from Spectris Instrument Systems, Shanghai, China. The non-immersion backscattering (NIBS) technology was used, which allowed for a wider range of sample concentration and particle size application. The viscosity of the polished slurry was tested with a NDJ-8S viscometer with rotor #1, equipment from Shanghai Lixin Bonsey Instrument Technology Co., Ltd., Shanghai, China. UV absorbance of the polished slurry was measured using a UV-2700i UV-Vis spectrophotometer, equipment from Shimadzu Instruments Ltd., Kyoto, Japan.

The valence change of elements in fused silica during polishing was determined using K-Alpha XPS. It was immersed in a 15 wt% K_4_P_2_O_7_ solution for 24 h and dried before the test. The monochromatic X-ray source was realized by using Al Kα radiation (hv = 1486.6 eV) with a spot diameter of 400 μm. The passage energies of the X-ray gun in the full spectrum were 12 kV, 6 mA, and 100 eV. The passage energies of the fine spectrum were 12 kV, 6 mA, and 50 eV. All spectra were calibrated using C1s at 284.8 eV. The XPS data were analyzed using Avantage 5.9 software. Changes in peak functional group vibrations before and after cyclic polishing of the polished slurry were detected using FTIR.

## 3. Results and Discussion

### 3.1. Quartz Glass Orthogonal Experiments

Figure 1 shows the quartz glass’s 3D contour and surface morphology before and after grinding. Before grinding, the surface of the quartz glass sample showed extensive scratches. After 5 min of grinding, many surface scratches and brittle fractures were removed. The S_a_ was reduced from 483.20 nm to 22.25 nm in the measurement area of 100 × 100 μm^2^. After the grinding process, the surface mainly consisted of ductile scratches, while no significant brittle chipping pits were found. As a result, the sub-surface damage depth was reduced greatly, and the polishing time for the subsequent CMP was minimized.

The shape of the abrasive is an essential factor affecting polishing quality. Figure 2 shows the SEM images of CeO_2_ and LaOF abrasives [20] and the novel polishing slurry configuration process. CeO_2_ abrasives were spherical, while the LaOF abrasives are hexahedral [21,22] in morphology. It was considered that more cutting edges or polishing sites were beneficial for the MRR. However, scratches induced on the surface of the dispersion of slurry or size distribution were not uniform. The particle size of both abrasives was 100 nm, and the size distribution for both abrasives was consistent. Therefore, both abrasives induced material removal due to their similar size.

To study how different parameters influenced the polishing performance and obtain the best polishing parameter combination, orthogonal experiments for six influencing factors, namely: abrasive concentration, polishing slurry pH, dispersant concentration, pressure, speed, and flow rate of polishing [23,24,25,26,27,28,29] were conducted. The first three parameters were for the chemical composition in the slurry, while the latter parameters were for the polishing process. It was expected that both chemical components and mechanical processing parameters would have an important effect on the polishing performance. However, there was also a coupling effect between them [30]. The specific data of the orthogonal experiments are shown in Table A1, the surface profile in Figure A1, and the optical morphology in Figure A2.

The surface of all groups has a surface roughness around or under 1 nm, much smaller than the surface roughness after grinding. However, some of the surfaces have defects, such as small scratches and pits, as shown in Figure 3a,b. This might be related to poor dispersion performance when the abrasive concentration was too high, or the dispersant concentration was too low. In such cases, secondary agglomeration was observed, and large particles were formed in the slurry resulting in scratches and pits. Moreover, the orange peel was another defect found in another group, as shown in Figure 3c,d, but the forming machine was different from the scratches. In CMP, if the concentration of the chemical composition was too high, and the chemical etching was too severe, the softened layer did not remove effectively. As a result, further etching into the substrate was initiated, resulting in a rough surface with defects such as the orange peel.

The experimental data on S_a_ and MRR were obtained from the above orthogonal experiments. Both the S_a_ and MRR results were considered when deriving the best combination of parameters using the extreme difference analysis [31]. In both cases, the $K_{\mathrm{jm}}$, $\bar{K_{\mathrm{jm}}}$ and $R_{j}$ were calculated. The order of these factors affecting the polishing and the superior level of each factor were obtained. Finally, by comparing and analyzing the influence of the two test indices, an optimal parameter combination was found in the extreme difference; $R_{j}$ of the jth column is expressed as:(2)$$R_{j\;}=\max\left(\bar{K_{j1}},\;\bar{K_{j2}},\;\cdots ,\;\bar{K_{\mathrm{jm}}}\right)-\min\;\left(\bar{K_{j1}},\;\bar{K_{j2}},\;\cdots ,\;\bar{K_{\mathrm{jm}}}\right)$$
where $K_{\mathrm{jm}}$ is the sum of the test indices corresponding to the mth level of the factor in the jth column. $\bar{K_{\mathrm{jm}}}$ is the mean of $K_{\mathrm{jm}}$. The primary and secondary order of the factors [32] can be judged by $R_{j}$. The data in Table A2 and Table A3 corresponded to the orthogonal experimental polar-difference analysis data for the S_a_ and MRR test metrics, respectively. The result showed that the order of influence of the factors on S_a_ was D > E > A > C > B > F. The order of influence on MRR was D > B > E > C > F > A. The factors D (pressure) and E (speed) were the most important factors for both the S_a_ and the MRR, which is perfectly normal for the polishing process since they were the two main factors that govern all polishing methods according to the Preston formulation.

Figure 4 shows the effect of each factor on S_a_ and MRR. The overall trend of S_a_ in Figure 4a decreased and then increased as the abrasive concentration increased. However, the MRR variation trend was the opposite. At the concentration of 0.5 wt%, S_a_ was 0.42 nm, and MRR was 181.34 nm/min. With an increased abrasive concentration, the number of abrasive grains involved in grinding at the same time increased. Hence, high spots on the quartz glass surface were effectively removed, achieving the goal of reducing S_a_ and increasing the MRR. However, as the concentration continued to increase, too many abrasive grains entered the gap between the polishing pad and the sample. The average load on each abrasive grit became smaller, and as a result, the MRR decreased. Meanwhile, more abrasives meant a higher polishing heat, and the chemical corrosion effect became greater. As a result, S_a_ increased accordingly. After that, as the concentration continued to grow, MRR increased again. In this circumstance, the secondary regrouping of abrasives started to take effect as the abrasive concentration was too much for uniform dispersion, forming large particles. These large particles entered the polishing zone and created significant scratches and other defects, while the load on these large particles was much higher than others. As a result, the MRR increased, and the surface became rougher.

In terms of the influence of the chemical composition, as shown in Figure 4b, the S_a_ first decreased, then increased, and finally decreased as the pH increased, while the MRR had an opposite variation trend. When the pH was 9.5, the S_a_ was 0.45 nm, and the MRR was 194.20 nm/min. When the pH value increased, the alkaline environment benefited the chemical etching process during the CMP. However, when the pH increased past 10.2, the chemical etching effect was too great, and the softened layer could not be removed effectively. As a result, the S_a_ became much higher.

Figure 4d,e show that higher polishing pressure and speed were beneficial for the polishing performance in the selected range in this work since the S_a_ in both factors continued to decrease. However, it was also observed that the MRR experienced a quick drop when the pressure and the speed were at a certain value. In Figure 4f, it was observed that the flow rate of slurry had little influence on both the MRR and S_a_. The MRR did experience a large drop during the process, but after that, the MRR was almost at the same level even though the flow rate doubled. It was observed that a larger flow rate effectively removed the polishing chips from the polishing zone and supplied fresh and new abrasives into the polishing zone. However, the effect on MRR or S_a_ was minimal. On the other hand, it was considered that the new slurry at room temperature could also carry away some of the polishing heat, resulting in a drop in the CMP efficiency and thus the decreasing MRR.

The analysis above showed that the optimal level for S_a_ was A_2_ B_2_ C_3_ D_5_ E_5_ F_5_. However, when both levels were considered, the most optimal combination of the two tests was A_2_ B_2_ C_3_ D_5_ E_5_ F_1_, which was a 0.5 wt% abrasive concentration, pH 9.5, 0.5 wt% dispersant concentration, 40 kPa polishing pressure, 140 rpm polishing speed, and 2 mL/min polishing slurry flow rate. Repeated experiments at the same conditions were also performed on the optimum combination from the orthogonal experimental analysis to verify the results. Figure 5 shows the S_a_ and morphology after polishing with the optimal combination. Polished surfaces were smooth and free of surface flaws such as scratches and pits, and the S_a_ was as low as 0.23 nm.

Cross-sectional TEM images of the quartz glass before and after polishing are shown in Figure 6. Before CMP, the quartz glass surface layer exhibited cracks with a damaged layer of approximately 172.3 nm (Figure 6a). These cracks were irreversible plastic flow formed by surface compressive stresses during grinding. After the CMP, the microscopic cracks in the surface layer were completely removed, leaving only a dense-lattice layer with a thickness of 4.5 nm (Figure 6b). It was noticed that the TEM images for amorphous materials were rarely taken; therefore, it was still unknown what this layer was. From the TEM image, it was observed that there was no significant difference in the structure between the thin layer and the base material, which indicated that it was a structural compression deformation. It is well known that the open network of fused silica can temporarily store energy through network contraction, called densification. Such behavior cannot recover spontaneously but can be reversed by annealing to 0.8 *T*_g_ (glassification temperature). Another possibility is the plastic flow of the material on the surface was due to shear deformation, which is also a source of plastic deformation but cannot be recovered. It is the first time such a structure was observed under TEM. Future studies should focus on the forming mechanism of such a layer during CMP. Anyhow, no cracks were found on the surface after polishing, and the subsurface damage layer was greatly reduced, which showed the capability of CMP.

### 3.2. Polishing Slurry Dispersibility Experiment

The dispersibility of the polishing slurry is an important factor affecting polishing quality. Orthogonally influencing factors such as the polishing slurry concentration, pH, and dispersant concentration indicate slurry dispersal. With good dispersion performance, a more uniform particle size can be found in the slurry, and henceforth a lower S_a_ after polishing and higher MRR can be achieved. A particle size, potential, UV absorbance, and viscosity analysis was carried out for the polishing slurry with the optimum combination configuration following the orthogonal experiment to study the influence of dispersion performance [33,34,35,36,37,38]. The results are shown in Figure 7. The viscosity results are shown in Table 2. The results from the particle size analyzer show that the average particle size of the original polishing slurry was 460.3 nm. D50, the particle size median, was 207.6 nm, and D90 was 266.9 nm, meaning that 90% of the particles were below 266.9 nm. After dispersion, the average particle size of the polished slurry was 173.9 nm, almost one-third that of the original polishing slurry. The D50 was 157.9 nm, and D90 was 243.4 nm. A total of 10% of the original slurry had a size between 266.9 nm and 460.3 nm, which showed that the agglomeration of the abrasive before dispersion was severe. The massive size difference between the largest abrasive was the main cause of defects on polished surfaces with the poorly dispersed slurry, where the polish pressure was not evenly distributed.

The zeta potential decreased from −37.53 mV to −22.65 mV, while the maximum UV absorbance decreased from 1.925 to 0.317. The zeta potential measured the strength of mutual repulsion or attraction between particles. The UV absorbance indicated the dispersion of particles suspended in solution, and its value is proportional to the number of particles dispersed. Thus, by comparing the above result of the two polishing slurries before and after dispersion processing, it was found that the slurry after dispersion had a much better dispersion rate and stability in solution. However, the viscosity increased from 1.13 mPa·S to 5.06 mPa·S due to the addition of the PAAS dispersing agent. PAAS is a high-molecular-weight anionic dispersant. In addition to forming an electrical double layer on the abrasive surface in solution to overcome van der Waals forces between abrasive materials, it formed a spatial resistance at the site in the polishing slurry through its special functional groups, further increasing the stability of the solution.

The polishing slurry of quartz glass was configured according to the above method, and the polishing process consisted of green and environment friendly ingredients. LaOF and CeO_2_ are light rare earth materials, which do not pollute the environment. SNLS [39,40,41,42] is an amino-acid-like anionic surfactant having good surface activity capability, excellent biodegradability, and corrosion resistance, and is widely used in cosmetics and other fields. PAAS is a high-molecular-weight anionic dispersant [43,44,45]. The compound is approved for use in food additives and has many advantages, such as excellent mechanical properties, stability of fractals, and so on. K_4_P_2_O_7_ serves as a pH control agent [46]. Its continuous hydrolysis in water maintains the pH stability of the solution. K_4_P_2_O_7_ is also an eco-friendly chelating agent that can be used as a remediation of soil drenches to remove heavy metals from the soil. After polishing, only DI water and compressed air are used to clean the samples. Therefore, the entire process and polishing slurry product are green.

### 3.3. Investigated the Mechanical Chemical Polishing Mechanism of Quartz Glass

The XPS data are shown in Figure 8, and to understand the CMP mechanism of quartz glass, pieces of polished samples were immersed in a solution of K_4_P_2_O_7_ with a concentration of 5 wt% for 24 h.

The first reaction in the solution was the hydrolysis reaction of K_4_P_2_O_7_, which formed an alkali polishing environment. Because K_4_P_2_O_7_ is a strong base and a weak acid salt, it is a conjugate base of the pyrophosphate of tetradecanoic acid. Therefore, without considering K^+^, the form of ions present in its aqueous solution was theoretically the same as the form of ions present in pyrophosphate in an aqueous solution. Pyrophosphate [46] has an ionization constant of $K_{a1\;}$ = 7.5 × 10^−1^, $K_{a2\;}$ = 6.2 × 10^−2^, $K_{a3\;}$= 1.7 × 10^−6^, $K_{a4\;}$ = 6 × 10^−9^. From Equation (3), its constant dissociation ${\mathrm{pK}}_{a}$ was deduced.
(3)$${\mathrm{pK}}_{a}=-{\mathrm{lgK}}_{a\;}$$
where ${\mathrm{pK}}_{a1}\;$ = 0.1249, ${\mathrm{pK}}_{a2}$ = 1.2707, ${\mathrm{pK}}_{a3}$ = 5.769, ${\mathrm{pK}}_{a4}$ = 8.221. From the Henderson Hasselbach Equation (4), one can then derive the shape of its ion distribution in the solution [47]. The results are given in Figure 9a, where $\left[A^{-}\right]$ and [HA] are the corresponding substances’ concentrations, respectively.
(4)$${\mathrm{pH}=\mathrm{pK}}_{a}+\mathrm{lg}\frac{\left[A^{-}\right]}{\left[\mathrm{HA}\right]}$$

K_4_P_2_O_7_ in the solution was completely hydrolyzed at pH 9.5. This corroborated with the XPS data by the presence of large quantities of P_2_O_7_^4−^ and K^+^ in the solution. The energy difference of the orbital spin splitting peaks (2p_3/2_ and 2p_1/2_) in the P 2p spectrum was about 0.9 eV. Based on the energy positions of the P 2p spectral peaks and the database, it was judged that P is present as P_2_O_7_^4−^, and the binding energies of P 2p_3/2_ and P 2p_1/2_ are 133.0 eV and 133.9 eV, respectively [48]. The peaks of 2p_3/2_ and 2p_1/2_ of K^+^ in the samples correspond to 293.15 eV and 296.3 eV, respectively [49]. O 1 s could be divided into metal oxides, K_2_SiO_3_, SiO_2,_ and H_2_O components. The binding energies were 530.5, 532.15, 532.8, and 534.3 eV, respectively [50,51,52], where K_2_SiO_3_ is related to the K_4_P_2_O_7_ in the sample. In comparison, Si 2p corresponded to the peaks of K_2_SiO_3_ and SiO_2_ with binding energies at 102.55 eV and 103.3 eV, respectively [53,54]. The mountains of K_2_SiO_3_ and SiO_2_ could be correlated with the results of O 1 s. SiO_3_^2−^ was generated by the chemical reaction in the alkaline environment generated by the hydrolysis of K_4_P_2_O_7_. The reaction mechanism for this polishing process is thus as follows.
(5)$${4K}^{+}{+P}_{2}O_{7}{}^{4-}{+H}_{2}O\to {\mathrm{HP}}_{2}O_{7}{}^{3-}{+\mathrm{OH}}^{-}{+4K}^{+}$$
(6)$${\mathrm{HP}}_{2}O_{7}{}^{3-}{+H}_{2}O\to H_{2}P_{2}O_{7}{}^{2-}{+\mathrm{OH}}^{-}$$
(7)$${\;H}_{2}P_{2}O_{7}{}^{2-}{+H}_{2}O\to H_{3}P_{2}O_{7}{}^{-}{+\mathrm{OH}}^{-}$$
(8)$$H_{3}P_{2}O_{7}{}^{-}{+H}_{2}O\to H_{4}P_{2}O_{7}{+\mathrm{OH}}^{-}$$
(9)$${\mathrm{SiO}}_{2}{+2\mathrm{OH}}^{-}\to H_{2}{O+\mathrm{SiO}}_{3}{}^{2-}$$

FTIR spectra were obtained to check the chelating effect of the dispersing agent on the polished products, as shown in Figure 9b. Before polishing, the 2849 cm^−1^ peak corresponded to the methylene group (CH_2_) vibration, attributed to SNLS and PAAS [39,43]. The band at 1667 cm^−1^ corresponded to the amino cation vibration, attributed to SNLS. The band at 1465 cm^−1^ corresponded to the methyl CH_3_ and methylene CH_2_ asymmetric bending vibration, attributed to the SNLS and PAAS. The band at 1286 cm^−1^ corresponded to the vibration of the pyrophosphate, assigned to the K_4_P_2_O_7_ [55]. The band at 1100 cm^−1^ corresponded to the C-O stretching vibration, attributed to SNLS and PAAS. After polishing, the corresponding assigned functional groups were observed at 2851 cm^−1^ corresponding to the methylene (CH_2_) vibration. The band at 1668 cm^−1^ corresponded to amino cations. The band at 1464 cm^−1^ corresponded to the methyl CH_3_ and the asymmetric flexural vibration of methylene CH_2_ group. The band at 1285 cm^−1^ corresponded to the pyrophosphate vibration. The band at 1101 cm^−1^ corresponded to the C-O stretching vibration. Firstly, ${\mathrm{OH}}^{-}$ generated by the hydrolysis of K_4_P_2_O_7_ in the solution provided an environment for polishing alkali. During this process, the quartz glass’s surface reacted with the slurry’s chemical components to generate Si-OH groups. SNLS and PAAS started to form complexes with the products through their special functional groups, leading to fluctuations in their infrared transmittance, as shown in Figure 10.

The mechanism of the quartz glass CMP was deduced from the XPS data and FTIR, as illustrated in Figure 11. First, K_4_P_2_O_7_ was hydrolyzed in the water, providing an alkaline polishing environment. The chemical interaction took place on the surface of SiO_2_ to produce silicates and Si-OH groups. The special functional groups of SNLS and PAAS in the polishing slurry then formed chelate products along with it. It was eventually mechanically removed by the CeO_2_ and LaOF rare earth abrasives in the polishing slurry. Since the hardness of both abrasives was lower than SiO_2_, the soft layer was removed without creating scratches on the polished surface. It was possible to obtain an ultra-smooth quartz glass surface. In the subsequent CMP, this process was repeated to complete the green and efficient low-damage chemical mechanical polishing of quartz glass by the dual action of chemical etching and mechanical removal.

## 4. Conclusions

This study used CeO_2_, LaOF, SNLS, PAAS, K_4_P2O_7,_ and DI water for the polishing slurry setup, which is composed entirely of green ingredients. Only DI water and compressed air were used in the polishing process for cleaning and drying the sample parts. The result of the polishing tests led to the following conclusions:Orthogonal testing was used to achieve the best polishing solution formulation for polishing quartz glass. The MRR of the polishing process was 530.52 nm/min, which was higher than the value reported previously. It was possible to achieve a surface roughness of 0.23 nm in the range of 50 × 50 μm^2^, which was less than the current 0.9 nm reported for commercial abrasives. All cracks and the subsurface damage layer were removed from the surface, leaving a 4.5 nm densified mesh structure.Particle size, zeta potential, UV absorbance, and viscosity were tested and analyzed for the polishing slurry before and after dispersion. The polishing slurry with the optimal combination from the orthogonal test had a more uniform particle size, a larger zeta potential, a higher UV absorbance, and a slightly higher viscosity than the untreated polishing slurry, showing that the dispersion is a very important factor for the polishing slurry.XPS and FTIR were used to analyze the polishing mechanism. It was discovered that K_4_P_2_O_7_ continually hydrolyzed in water to release the hydroxyl groups, forming an alkali-polishing environment. Then, SiO_2_ forms an attenuated layer of Si-OH groups. Due to their high electronegativity, anionic dispersants that are added to the SNLS and PAAS will then form chelate products with them via their polar headgroups. Finally, CeO_2_ and LaOF abrasives removed the softened layer to obtain quartz glass sample pieces with smooth surfaces.

## Figures and Tables

**Figure 1 materials-16-01148-f001:**
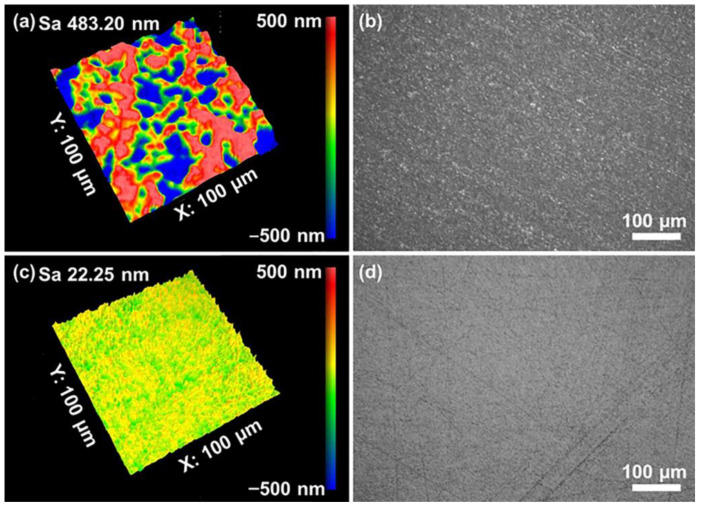
Surface morphology of quartz glass (**a**,**b**) before and (**c**,**d**) after grinding.

**Figure 2 materials-16-01148-f002:**
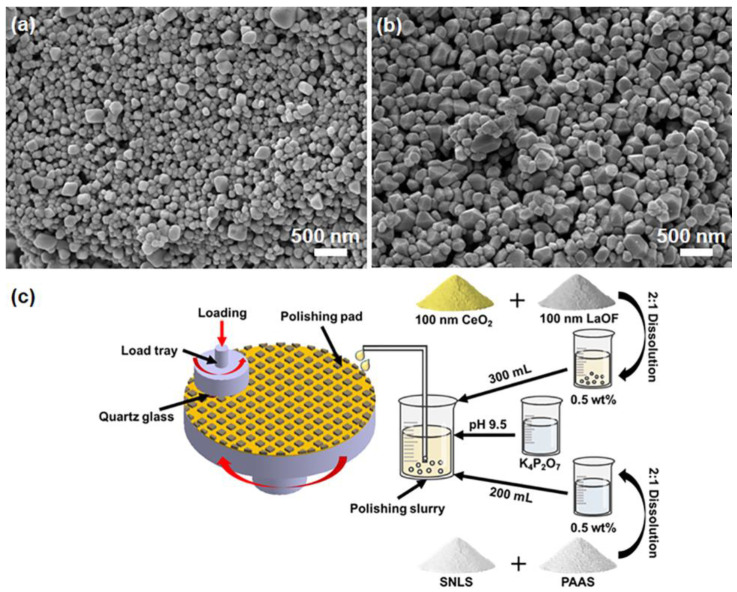
SEM image of (**a**) cerium oxide abrasive; (**b**) lanthanum fluoride oxide abrasive; (**c**) the configuration of the polishing slurry and the polishing process schematic.

**Figure 3 materials-16-01148-f003:**
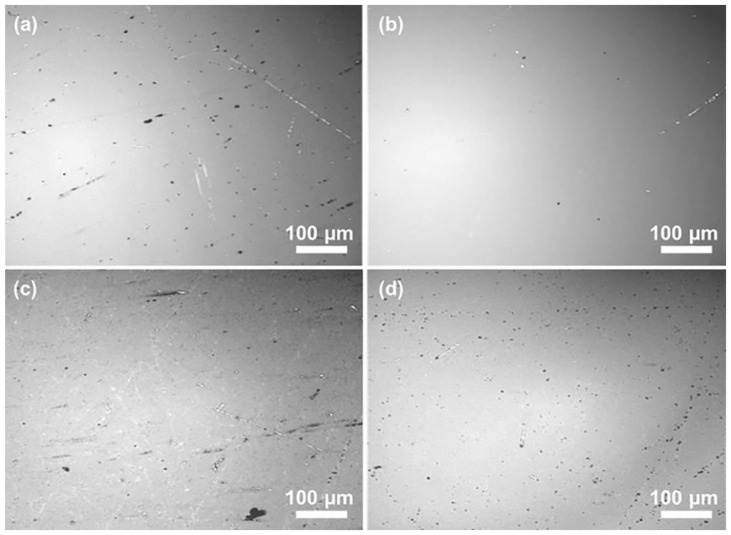
(**a**,**b**) Surface defects like scratches and pits; (**c**,**d**) Orange peel defects due to excessive corrosion.

**Figure 4 materials-16-01148-f004:**
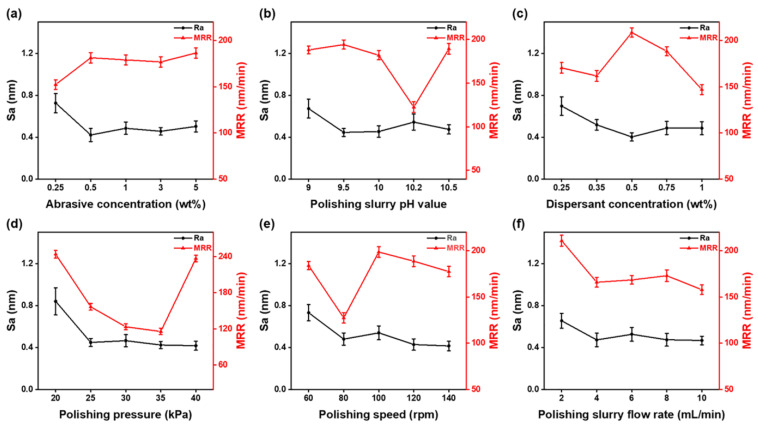
Effect of (**a**) abrasive concentration, (**b**) pH, (**c**) dispersant concentration, (**d**) polishing pressure, (**e**) polishing speed, and (**f**) polishing slurry flow rate on Sa and MRR in orthogonal experiments.

**Figure 5 materials-16-01148-f005:**
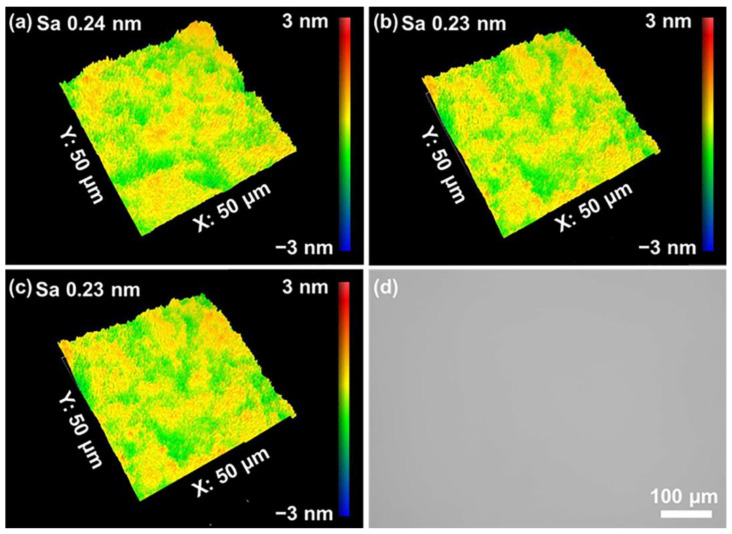
(**a**–**c**) The surface morphology and surface roughness on three random spots; and (**d**) an optical image of the polished sample.

**Figure 6 materials-16-01148-f006:**
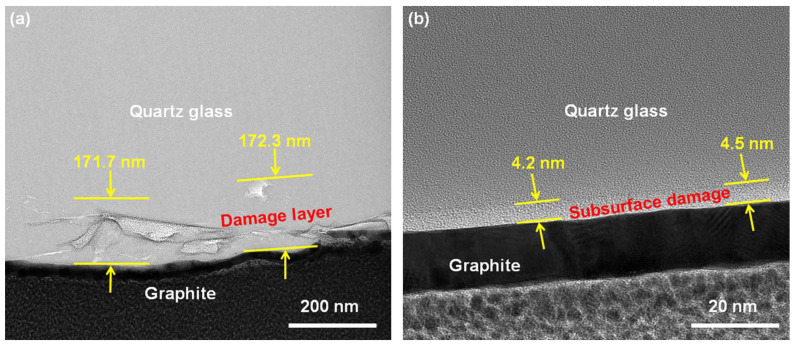
TEM cross-section of quartz glass (**a**) before polishing; (**b**) after polishing.

**Figure 7 materials-16-01148-f007:**
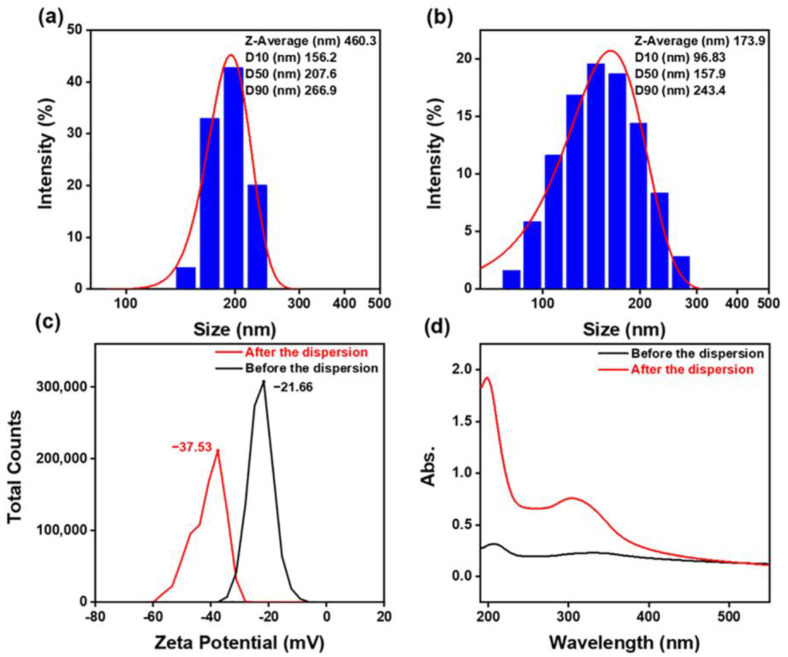
(**a**,**b**) Comparison of particle size of polishing slurry before and after dispersion; (**c**) comparison of zeta potential of polishing slurry before and after dispersion; (**d**) comparison of UV absorbance of polishing slurry before and after dispersion.

**Figure 8 materials-16-01148-f008:**
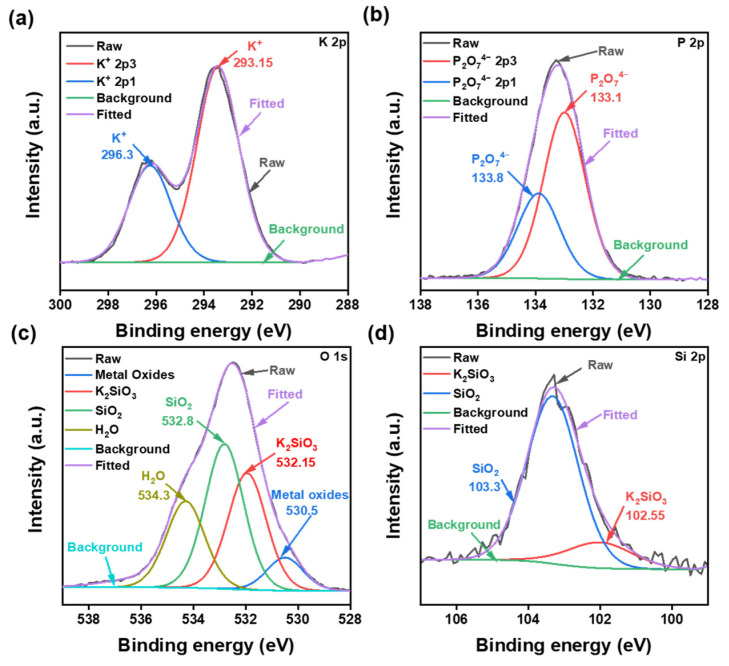
XPS fine spectra of quartz glass after immersion test. (**a**) K 2p, containing two peaks near 296.3 and 293.15 for K^+^; (**b**) P 2p, containing two peaks near 133.1 and 133.8 for P_2_O_7_^4−^; (**c**) O 1s, containing four peaks near 532.8 for SiO_2_, 532.15 for K_2_SiO_3_, 534.3 for H_2_O and 530.5 for Metal oxides; (**d**) Si 2p, containing two peaks of 103.3 for SiO_2_ and 102.55 for K_2_SiO_3_.

**Figure 9 materials-16-01148-f009:**
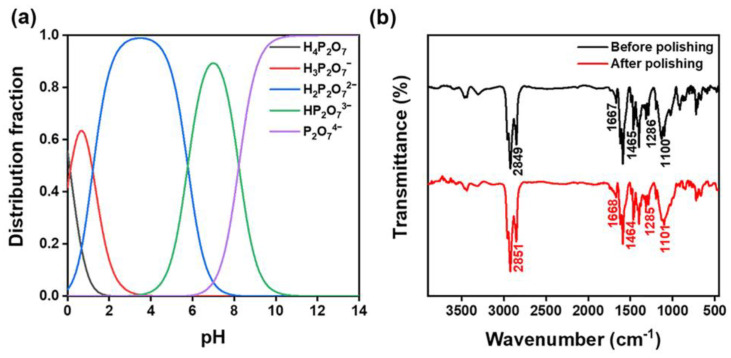
(**a**) Potassium pyrophosphate ion morphology distribution, (**b**) FTIR result of the slurry before and after polishing.

**Figure 10 materials-16-01148-f010:**
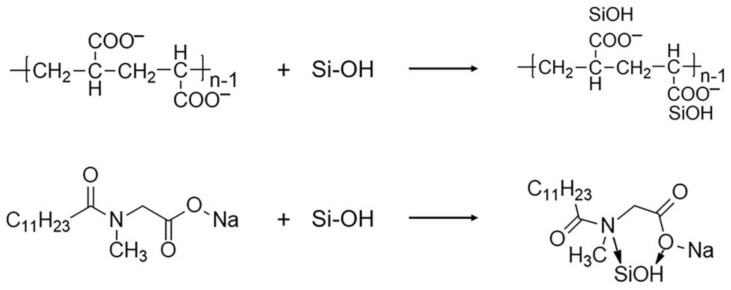
Dispersant chelation reaction with softened layer.

**Figure 11 materials-16-01148-f011:**
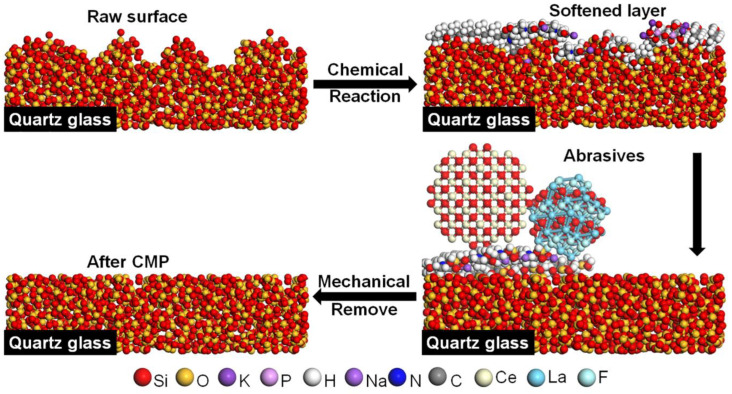
Chemical mechanical polishing mechanism of the slurry in this work.

**Table 1 materials-16-01148-t001:** L5^6^ orthogonal experiment factor level table.

No.	Factors	Levels				
		1	2	3	3	5
A	Abrasive concentration (wt%)	0.25	0.5	1	3	5
B	Polishing solution pH value	9	9.5	10	10.2	10.5
C	Dispersant concentration (wt%)	0.25	0.35	0.5	0.75	1
D	Polishing pressure (kPa)	20	25	30	35	40
E	Polishing speed (rpm)	60	80	100	120	140
F	Polishing fluid flow rate (mL/min)	2	4	6	8	10

**Table 2 materials-16-01148-t002:** Comparison of the viscosity of the polishing slurry before and after dispersion.

Index	Before the Dispersion	After the Dispersion
Viscosity (mPa·S)	1.13	5.06
Rotor (#)	0	0
Rotational speed (rpm)	60	60
Torque (%)	11.3	50.6

## Data Availability

Not applicable.

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
