# Peer review of "Dispersion and Polishing Mechanism of a Novel CeO2-LaOF-Based Chemical Mechanical Polishing Slurry for Quartz Glass"

_materials, 2023, doi:10.3390/ma16031148_

Round 1

Reviewer 1 Report

The paper "Study Dispersion and Polishing Mechanism of a Novel CeO2-LaOF-based Chemical Mechanical Polishing Slurry for Quartz Glass" contains useful information for glass, glass ceramics, and other optical materials. The paper is well-written and simple to read. The manuscript's organisation, figures, and tables are appropriate, and the conclusions are primarily supported by experimental characterization. For these reasons, I recommend that the manuscript be published following minor revisions, taking into account the following comments and suggestions.

1.       The authors should specify the type of grinding media used in a ball mill technique in section 2.2.

2.       Although Figure 3 depicts the surface defects that are significant results in this work, the images provided are not clear. Please provide higher-resolution images.

3. Check Figure 4 to confirm whether Ra or Sa should be used in the figure axes and legends

Author Response

Dear reviewer,

Thank you very much for your review comment. We have revied the manuscript according to your comments. Shown below is the detailed information on how I responded to the comments, highlighting the revisions made in the manuscript. When reading the information below, please note:

Text in italic style: the comments

Text in regular style: my responses to the comments.

Changes are highlighted in red color in the revised manuscript.

REVIEWER REPORT(S):

Comments and Suggestions for Authors:

The paper "Study Dispersion and Polishing Mechanism of a Novel CeO2-LaOF-based Chemical Mechanical Polishing Slurry for Quartz Glass" contains useful information for glass, glass ceramics, and other optical materials. The paper is well-written and simple to read. The manuscript's organisation, figures, and tables are appropriate, and the conclusions are primarily supported by experimental characterization. For these reasons, I recommend that the manuscript be published following minor revisions, taking into account the following comments and suggestions.

  1. The authors should specify the type of grinding media used in a ball mill technique in section 2.2.
    We have added the type of grinding media, and the material is zirconium ball.
  2. Although Figure 3 depicts the surface defects that are significant results in this work, the images provided are not clear. Please provide higher-resolution images.
    Thank you for your comment. We have replaced the optical images with clearer ones to better show the surface defects.
  3. Check Figure 4 to confirm whether Ra or Sa should be used in the figure axes and legends. 
    Thank you for pointing that out, we have replaced all the y-axes in Figure 4 with Sa, which corresponds to the content in the text.

At last, we want to thank the reviewer for all the helpful comments, which has definitely improved the quality of this work.

Yours sincerely,

Zhenyu

Reviewer 2 Report

The authors present a study using CeO2, LaOF, SNLS, PAAS, K4P2O7, and DI water for the polishing slurry, which consists entirely of green components. It is a very interesting concept to polish the surface of quartz glass.

The paper is clearly structured and well written. Only two minor revisions would be helpfull:

1) All measures of the surface roughness Sa and the material removal rate MRR were given with 3 digits after the comma, which means with pm resolution. I think it would be more trustable with at least 2 digits.

2) In Fig.4 the authors should exchange the y-axis label from Ra into Sa.

Overall thanks for this nice and helpful paper.

Author Response

Dear reviewer,

Thank you very much for your review comment. We have revied the manuscript according to your comments. Shown below is the detailed information on how I responded to the comments, highlighting the revisions made in the manuscript. When reading the information below, please note:

Text in italic style: the comments

Text in regular style: my responses to the comments.

Changes are highlighted in red color in the revised manuscript.

REVIEWER REPORT(S):

Comments and Suggestions for Authors:

The authors present a study using CeO2, LaOF, SNLS, PAAS, K4P2O7, and DI water for the polishing slurry, which consists entirely of green components. It is a very interesting concept to polish the surface of quartz glass.

The paper is clearly structured and well written. Only two minor revisions would be helpful:

  1. All measures of the surface roughness Saand the material removal rate MRR were given with 3 digits after the comma, which means with pm resolution. I think it would be more trustable with at least 2 digits.
    Thank you for your comment. We have changed all the values of Sa, MRR to two decimal places to increase the credibility of the result.
  2. In Fig.4 the authors should exchange the y-axis label from Ra into Sa.
    Thank you for pointing that out, we have replaced all the y-axes in Figure 4 with Sa, which corresponds to the content in the text.

At last, we want to thank the reviewer for all the helpful comments, which has definitely improved the quality of this work.

Yours sincerely,

Zhenyu